

# Sequence variant analysis of RNA sequences in severe equine asthma

Laurence Tessier[1,2], Olivier Côté[1,3] and Dorothee Bienzle[1]

[1] Department of Pathobiology, University of Guelph, Guelph, ON, Canada
[2] BenchSci, Toronto, ON, Canada
[3] BioAssay Works, Ijamsville, MD, USA

## ABSTRACT

**Background:** Severe equine asthma is a chronic inflammatory disease of the lung in horses similar to low-Th2 late-onset asthma in humans. This study aimed to determine the utility of RNA-Seq to call gene sequence variants, and to identify sequence variants of potential relevance to the pathogenesis of asthma.

**Methods:** RNA-Seq data were generated from endobronchial biopsies collected from six asthmatic and seven non-asthmatic horses before and after challenge (26 samples total). Sequences were aligned to the equine genome with Spliced Transcripts Alignment to Reference software. Read preparation for sequence variant calling was performed with Picard tools and Genome Analysis Toolkit (GATK). Sequence variants were called and filtered using GATK and Ensembl Variant Effect Predictor (VEP) tools, and two RNA-Seq predicted sequence variants were investigated with both PCR and Sanger sequencing. Supplementary analysis of novel sequence variant selection with VEP was based on a score of <0.01 predicted with Sorting Intolerant from Tolerant software, missense nature, location within the protein coding sequence and presence in all asthmatic individuals. For select variants, effect on protein function was assessed with Polymorphism Phenotyping 2 and screening for non-acceptable polymorphism 2 software. Sequences were aligned and 3D protein structures predicted with Geneious software. Difference in allele frequency between the groups was assessed using a Pearson's Chi-squared test with Yates' continuity correction, and difference in genotype frequency was calculated using the Fisher's exact test for count data.

**Results:** RNA-Seq variant calling and filtering correctly identified substitution variants in *PACRG* and *RTTN*. Sanger sequencing confirmed that the *PACRG* substitution was appropriately identified in all 26 samples while the *RTTN* substitution was identified correctly in 24 of 26 samples. These variants of uncertain significance had substitutions that were predicted to result in loss of function and to be non-neutral. Amino acid substitutions projected no change of hydrophobicity and isoelectric point in PACRG, and a change in both for RTTN. For *PACRG*, no difference in allele frequency between the two groups was detected but a higher proportion of asthmatic horses had the altered *RTTN* allele compared to non-asthmatic animals.

**Discussion:** RNA-Seq was sensitive and specific for calling gene sequence variants in this disease model. Even moderate coverage (<10–20 counts per million) yielded correct identification in 92% of samples, suggesting RNA-Seq may be suitable to detect sequence variants in low coverage samples. The impact of amino acid

Corresponding author
Dorothee Bienzle,
dbienzle@uoguelph.ca

alterations in PACRG and RTTN proteins, and possible association of the sequence variants with asthma, is of uncertain significance, but their role in ciliary function may be of future interest.

## INTRODUCTION

Severe equine asthma (recurrent airway obstruction, heaves) is a chronic inflammatory lung disease caused by inhalation of environmental dust and microbial components (*Couëtil et al., 2016*). Exacerbation of the disease triggers excessive mucus production, cough, neutrophilic airway inflammation, bronchial hyperreactivity, and bronchospasm. Recurrent exacerbations induce smooth muscle hyperplasia and hypertrophy, fibrosis and eventual irreversible airway remodeling (*Pirie, 2014*; *Martinez & Vercelli, 2013*; *Vargas et al., 2016*; *Setlakwe et al., 2014*).

Asthma in humans is recognized to be a heterogeneous disease that is classified considering genetic, molecular and clinical features (*Wenzel, 2012*; *Skloot, 2016*). Severe equine asthma is most similar to human severe, late-onset asthma characterized by absence of Th2 cytokines, and presence of neutrophilic inflammation and bronchial neutrophil chemokines (*Wenzel, 2012*; *Ilmarinen, Tuomisto & Kankaanranta, 2015*). Severely asthmatic horses do not have a hypersensitivity response (*Pirie, 2014*) and efforts to associate equine asthma with a Th2 cytokine profile have yielded inconsistent or inconclusive results (*Lavoie et al., 2001*; *Joubert, Silversides & Lavoie, 2001*; *Klukowska-Rötzler et al., 2012a*; *Ainsworth et al., 2003*, *2007*, *2009*; *Debrue et al., 2005*; *Lavoie-Lamoureux et al., 2010*; *Padoan et al., 2013*). Mechanisms leading to the development of both severe equine asthma and late-onset low-Th2 severe asthma in humans remain largely undefined.

Interactions between genes and environmental factors have been recognized to contribute to development of equine asthma for many years (*Marti et al., 1991*). Genetic factors likely reside in multiple gene sequence variants, and may be influenced by age and sex (*Marti et al., 1991*; *Couëtil & Ward, 2003*; *Ramseyer et al., 2007*). Several susceptibility sequence variants, haplotypes and regions have been associated with human asthma (*Moffatt et al., 2007*; *Choudhry et al., 2008*; *Galanter et al., 2008*; *Tavendale et al., 2008*; *Madore et al., 2008*; *Hancock et al., 2009*; *Himes et al., 2009*; *Bisgaard et al., 2009*; *Wu et al., 2009*; *Li et al., 2010*; *Sleiman et al., 2010*; *Moffatt et al., 2010*; *Ferreira et al., 2011*; *Torgerson et al., 2011*) but no specific markers have been identified in the late-onset low-Th2 sub-phenotype (*Wenzel, 2012*; *Ilmarinen, Tuomisto & Kankaanranta, 2015*). Similarly, genetic markers of equine severe asthma were identified in certain families, but were not significantly associated across different families and genetic backgrounds (*Ramseyer et al., 2007*; *Swinburne et al., 2009*; *Klukowska-Rötzler et al., 2012b*; *Jost et al., 2007*; *Klukowska-Rötzler, Gerber & Leeb, 2012*).

RNA-Seq is a promising approach for calling sequence variants concurrent with analysis of gene and allele-specific expression, alternative splicing, and pathways (*Guo et al., 2015*). In this study we investigated whether single nucleotide variant (SNV) detected by RNA-Seq were also present in Sanger-sequenced amplicons. We hypothesized that RNA-Seq would identify gene sequence variants with high accuracy.

## METHODS

### Animals and procedures

Animal and sample procedures were previously described (*Tessier et al., 2017*). Briefly, six asthmatic and seven non-asthmatic horses without signs of asthma exacerbation belonging to the institutional research herd (mean ages of 15 and 12 years, respectively, $P = 0.352$, unpaired *t*-test) were placed indoor in a dust-free environment. After 24 h, asthmatic horses were exposed to dusty hay until exacerbation (range 1–3 days, average 2.2 days), while non-asthmatic horses were exposed for 3 days. Before and after the dusty hay asthmatic challenge, physical examination, pulmonary function test, and bronchoalveolar lavage were performed, and endoscopic bronchial biopsies were collected from lung lobes contralateral between first and second samples. Samples from an additional four asthmatic and seven non-asthmatic horses were used for PCR-amplification of specific sequence variant regions and Sanger sequencing. All procedures were approved by the Institutional Animal Care Committee of the University of Guelph (protocol R10-031) and conducted in compliance with Canadian Council on Animal Care guidelines.

### RNA-Seq sample preparation and sequence alignment

RNA extraction, preparation, and sequencing procedures were as described previously (*Tessier et al., 2017*). In brief, total RNA was extracted from endobronchial biopsies (Qiagen, Toronto, ON, Canada) and tested for quality and concentration with the Bioanalyzer RNA Nanochip (Agilent, Mississauga, ON, Canada) and capillary electrophoresis. RNA-Seq library preparation (unstranded) and sequencing were performed using the Illumina TruSeq RNA sample preparation and appropriate sequencing protocols (Illumina, San Diego, CA, USA) at The Centre for Applied Genomics (Toronto, ON, Canada). Sequencing of 100-base paired-end reads was performed following the manufacturer's instructions on an Illumina HiSeq 2500 instrument.

FastQC software version 0.10.1 (bioinformatics.babraham.ac.uk/projects/fastqc/) was used to assess quality of raw reads, and alignment to the horse reference genome (*Wade et al., 2009*) (Ensembl v70) was performed with STAR version 2.4 (*Dobin et al., 2013*). Specifically, the STAR_pass2 alignment protocol was followed using the horse Ensembl version 70 GTF annotation file for first- and second-pass, and the junction SJ.tab file generated by STAR for the second-pass after non-canonical junctions were removed. Default settings were used except for: –runThreadN 8 –outFilterScoreMinOverLread 0.5 –outFilterMatchNminOverLread 0.5. Details and results for read alignment were previously described (*Tessier et al., 2017*).

## Sequence variant calling and filtering

Read processing, sequence variant calling, and initial filtering were performed following the Genome Analysis ToolKit (GATK) best practice guide for variant calling on RNA-Seq, except for the Indel realignment step considering the pass-2 STAR alignment initially performed. Initial read processing was first performed with Picard tools version 1.114 (broadinstitute.github.io/picard/) to add read groups and mark duplicates. Split n' Trim as well as base recalibration were performed using the GATK software version 3.2.2 (*McKenna et al., 2010*) and the *-T SplitNCigarReads*, *-rf ReassignOneMappingQuality*, *-RMQF 255*, *-RMQT 60*, and *-U ALLOW_N_CIGAR_READS* options.

The GATK variant calling and filtering workflow yielded 2,823 and 1,788 sequence variants present in all horses of the asthmatic group pre- and post-challenge, respectively (Fig. S1). Sequence variants were subsequently called using the Haplotype Caller function in GATK with the same genome annotation file used in the read alignment phase and the following options: *-recoverDanglingHeads*, *-dontUseSoftClippedBases*, *-stand_call_conf 20.0*, and *-stand_emit_conf 20.0* options. Resultant sequence variants were processed with the variant filtration function of GATK software and the following options to establish a confidence threshold of reported variants: *-window 35*, *-cluster 3*, *-filterName FS*, *-filter "FS > 30.0"*, *-filterName QD*, and *-filter "QD < 2.0."* Sequence variants were analyzed individually in each of 26 samples (six asthmatics and seven non-asthmatics, before and after asthmatic challenge).

## PCR

Primers for amplification of sequence variant regions from bronchial DNA were parkin co-regulated (*PACRG*) forward (5′-CTC TGA ACC TCC GAA ACC GAC-3′) and reverse (5′-CTC CTG GGA TAA CTC ACC ATT C-3′), and rotatin (*RTTN*) forward (5′-TCC TGA GTT GTA TCA AGA AGT G-3′) and reverse (5′-CCA GCC TGC AAT TCC TTT CT-3′). A Taq polymerase PCR kit (Invitrogen, Mississauga, ON, Canada) was used for PCR amplifications. Each reaction was performed in a 25 μL final volume, including five μL of 10× PCR buffer, 0.2 mM dNTPs, two mM $MgSO_4$, 0.3 μM of each primer, two U of Platinum Taq, and five μL (100 ng) of template DNA. PCR conditions for amplification were 3 min at 94 °C followed by 35 cycles of 94 °C for 45 s, 60 °C or 58 °C for 30 s for *PACRG* and *RTTN*, respectively, and 72 °C for 90 s, followed by final elongation for 10 min at 72 °C. A total of 20 μL of each PCR product was separated by electrophoresis in a 1% agarose gel stained with SYBR Safe (Invitrogen, Mississauga, ON, Canada). Amplicons of appropriate size were cut out and DNA extracted and purified (QIAquick; Qiagen, Toronto, ON, Canada). Extracted and purified PCR products were Sanger sequenced (Laboratory Services Division, Guelph, ON, Canada).

# RESULTS

## Sequence variant calling and filtering

The mean of the total number of RNA-Seq reads for all samples was 36252701.08, and the mean of uniquely mapped number of reads was 33127466.35. The number of

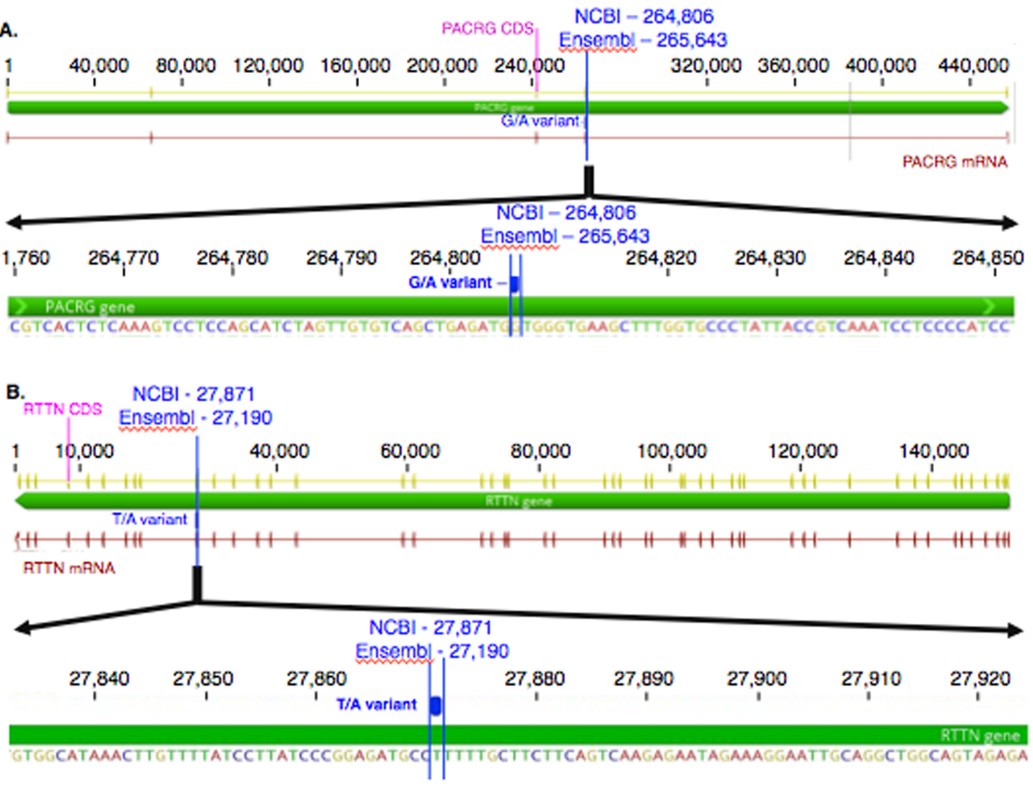

**Figure 1 Substitution variants in *PACRG* (A) and *RTTN* (B) genes.** Diagrams show position of gene (thick green line), mRNA (red line) and coding region (yellow line). The whole gene is outlined at the top and details of the nucleotide sequence around the substitution variant are shown at the bottom. Location of the substitution variant is indicated in blue for NCBI and Ensembl databases.

individual total reads, total mapped reads, uniquely mapped reads and multiple mapped reads is detailed elsewhere (39). The GATK workflow resulted in 2,823 and 1,788 sequence variants present specifically in the asthmatic group pre- and post-challenge, respectively (Fig. S1). Of these, 10 were missense substitution variants, coded for proteins and had Sorting Intolerant from Tolerant (SIFT) scores <0.01. Substitution variants in *PACRG* (Fig. 1A) and *RTTN* (Fig. 1B) were detected at higher proportion in asthmatic compared to non-asthmatic horses. A missense G/A substitution was detected in the coding sequence of *PACRG* at position 265,643 (Ensembl sequence ENSECAG00000014308)/264,806 (NCBI accession number 100050378) (Fig. 1A). A missense T/A substitution was detected in the coding sequence of *RTTN* at position 27,190 (Ensembl sequence ENSECAG00000009711)/27,871 (NCBI accession number 100052029) (Fig. 1B).

## Amino acid sequence alignment

In PACRG, the G/A substitution resulted in replacement of valine (V) for methionine (M) at position 182 (Fig. 2A). PACRG sequence alignment of wild type (WT) and altered proteins predicted changes from beta-strand to alpha-helix structure in the altered protein a few amino acids distant from the site of substitution (182) at positions 187 and 188

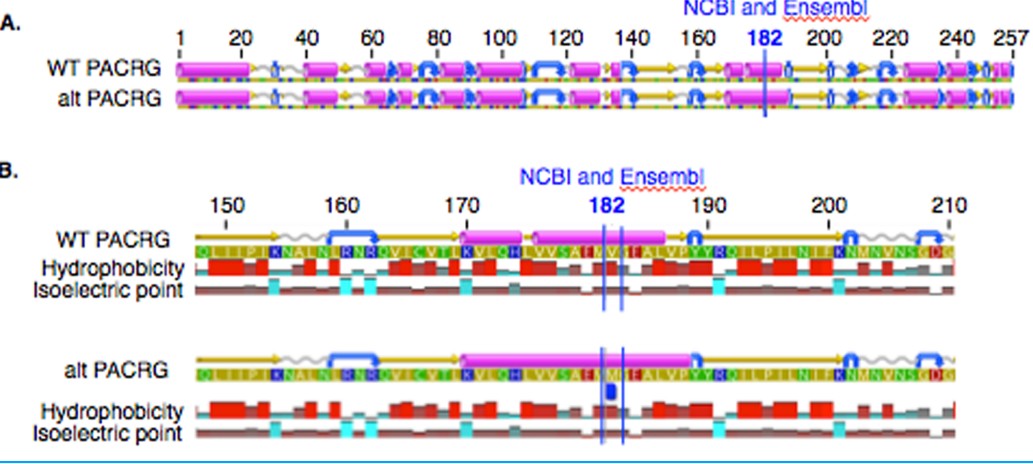

**Figure 2 Alignment of wild type (WT) and mutant (mut) PACRG proteins with associated predicted hydrophobicity and isoelectric point.** Low (A) and high magnification (B) of predicted protein changes. Replacement of methionine for valine at position 182 changes a beta strand to an alpha helix at position 187 and 188. Alpha helices (pink), coils (gray line), turns (blue arrows), and beta strands (yellow arrows).

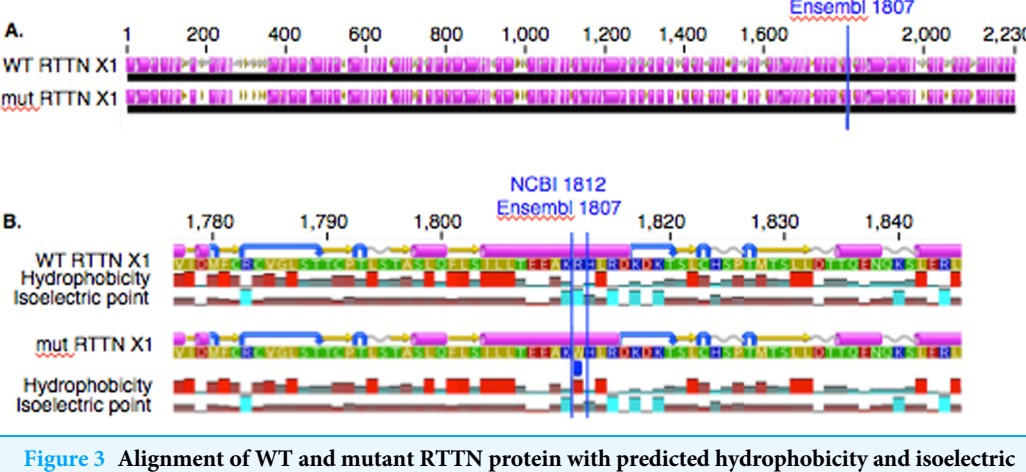

**Figure 3 Alignment of WT and mutant RTTN protein with predicted hydrophobicity and isoelectric point.** Low (A) and high magnification (B) of predicted protein changes. Replacement of arginine with tryptophan at position 1,812 changes alpha helix to beta strand at position 1,816. Alpha helices (pink), coils (gray line), turns (blue arrows), and beta strands (yellow arrows).

(Fig. 2B). Hydrophobicity and isoelectric point were expected to remain similar despite the substitution (Geneious). In the RTTN sequence alignment, T/A substitution resulted in replacement of arginine (R) with tryptophan (W) at position 1,807 of the ENSECAT00000010304 protein isoform (Ensembl sequence, corresponding to position 1,812 of isoform X1 in NCBI [XP_001493238]) in NCBI sequence) (Fig. 3A). Sequence alignment of WT and altered proteins indicated a change from alpha helix to beta strand structure near the site of substitution (bp 1,807) at position 1,816 (Fig. 3B). In addition, increased hydrophobicity and decreased pI were projected at the site of substitution (1,807) in the altered compared to the WT protein.

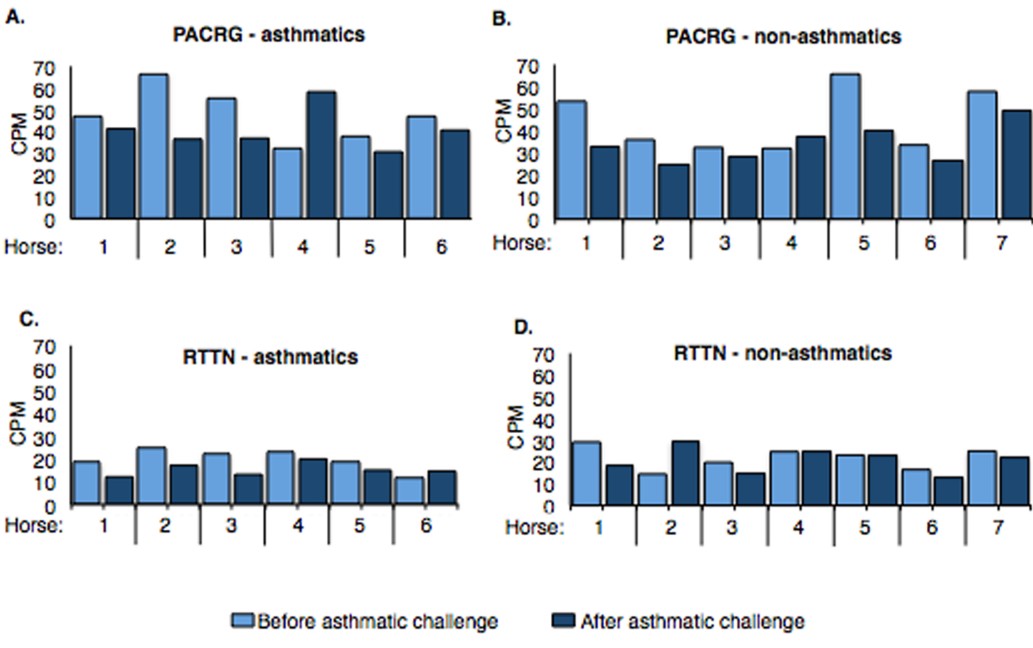

**Figure 4 Expression of PACRG (A, B) and RTTN (C, D) in asthmatic and non-asthmatic horses in counts-per-million (CPM; *y*-axis) pre- and post-challenge.** PACRG expression varied from ~30.7 to 66.3 CPM in asthmatic horses (A) and ~25.1 to 65.6 CPM on non-asthmatic horses (B), while RTTN expression varied from ~11.5 to 24.9 CPM and ~13.1 to 29.9 CPM in asthmatic (C) and non-asthmatic (D) horses, respectively.

## Expression of *PACRG* and *RTTN*

Counts per million (cpm) for *PACRG* ranged from ~30.7 to 66.3 (mean = 44.17) in asthmatic horses (Fig. 4A) and ~25.1 to 65.6 (mean = 39.31) in non-asthmatic horses (Fig. 4B), while expression of *RTTN* varied from ~11.5 to 24.9 (mean = 17.61) and ~13.1 to 29.9 (mean = 21.46) in asthmatic (Fig. 4C) and non-asthmatic (Fig. 4D) horses, respectively.

## Confirmation of RNA-Seq with DNA Sanger sequencing

The *PACRG* substitution variants identified by RNA-Seq were confirmed on DNA with Sanger sequencing of PCR amplicons in 10 asthmatic (Fig. 5A) and 14 non-asthmatic (Fig. 5B) horses. In the asthmatic group, four horses were heterozygous [A/G] and two were homozygous with altered alleles [A/A] (Fig. 5A). In the non-asthmatic group, one horse was heterozygous with alleles [A/G], three horses were homozygous WT [G/G] and three horses were homozygous altered [A/A] (Fig. 5B). DNA was not available to confirm the genotype of horse 1. Hence, all genotypes derived from RNA-Seq were identical to Sanger sequencing results.

For *RTTN*, results of RNA-Seq and Sanger sequencing were very similar (Fig. 6). Four asthmatic horses were heterozygous [A/T] and two were homozygous altered [A/A] (Fig. 6A). In the non-asthmatic group, horse 5 was heterozygous [A/T] and four horses were identified as homozygous WT [T/T]. However, horse 1 was identified as homozygous before and heterozygous after the asthmatic challenge, while horse 4 was identified as heterozygous before and homozygous after the asthmatic challenge. Based on Sanger

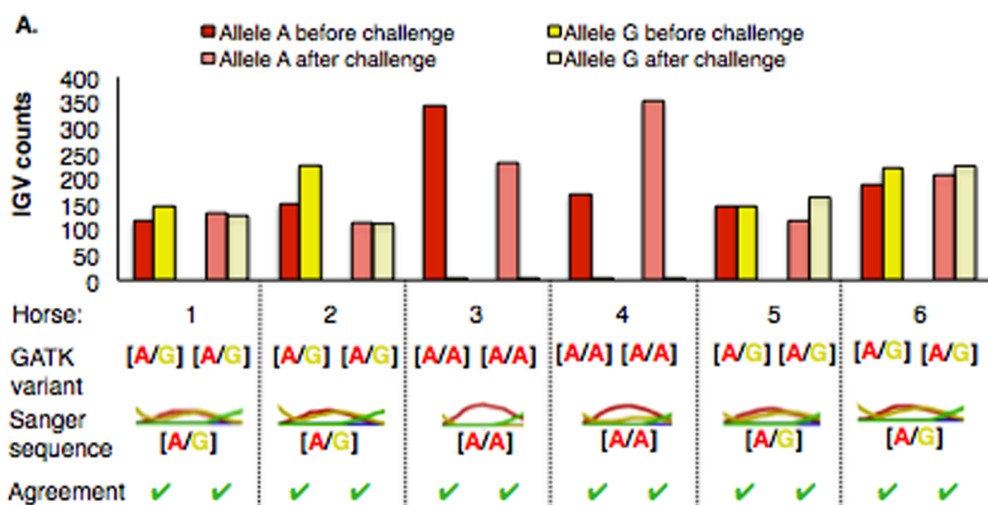

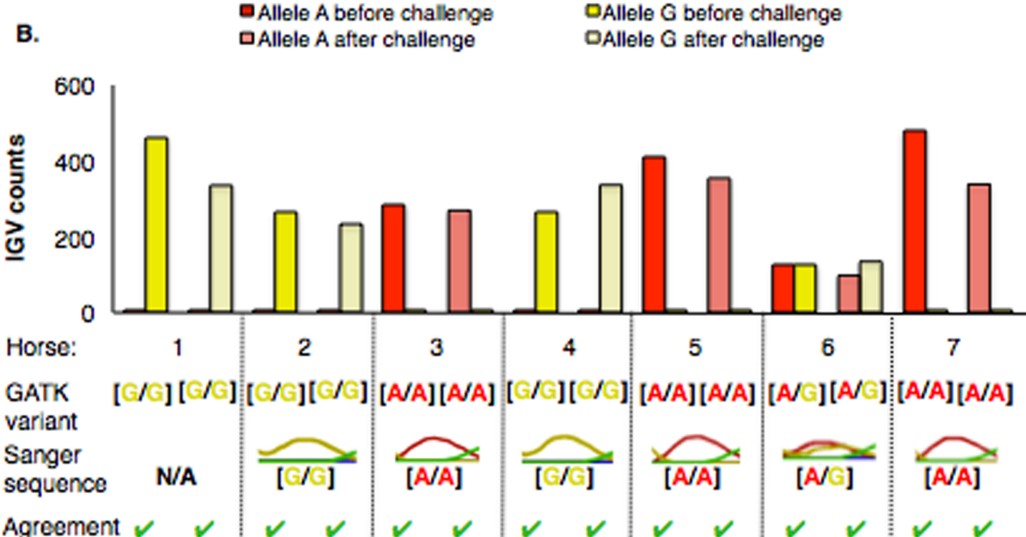

**Figure 5 Comparison of GATK variant calls and Sanger sequencing results for *PACRG* in asthmatic (A) and non-asthmatic (B) horses.** For both groups, the bar graph indicates the IGV count for each allele (A-red, G-yellow), horse and condition. Below the bar graph is the GATK variant call, the electropherogram of the Sanger sequence, and agreement. (A) Four asthmatic horses (1, 2, 5, 6) had heterozygous alleles [A/G] and two (3 and 4) were homozygous for the mutant allele [A/A]. (B) In non-asthmatic horses, one horse (6) had heterozygous alleles [A/G], three horses (1, 2, and 4) were homozygous for the wild-type allele [G/G] and three horses (3, 5, and 7) were homozygous for the mutant allele [A/A]. All genotypes were consistent across horses and methods. DNA was not available for non-asthmatic horse 1.

sequencing the genotype of horse 4 was homozygous. DNA was not available to confirm the genotype of horse 1. In all asthmatic horses, genotypes were consistent between pre- and post-challenge and sequencing methods.

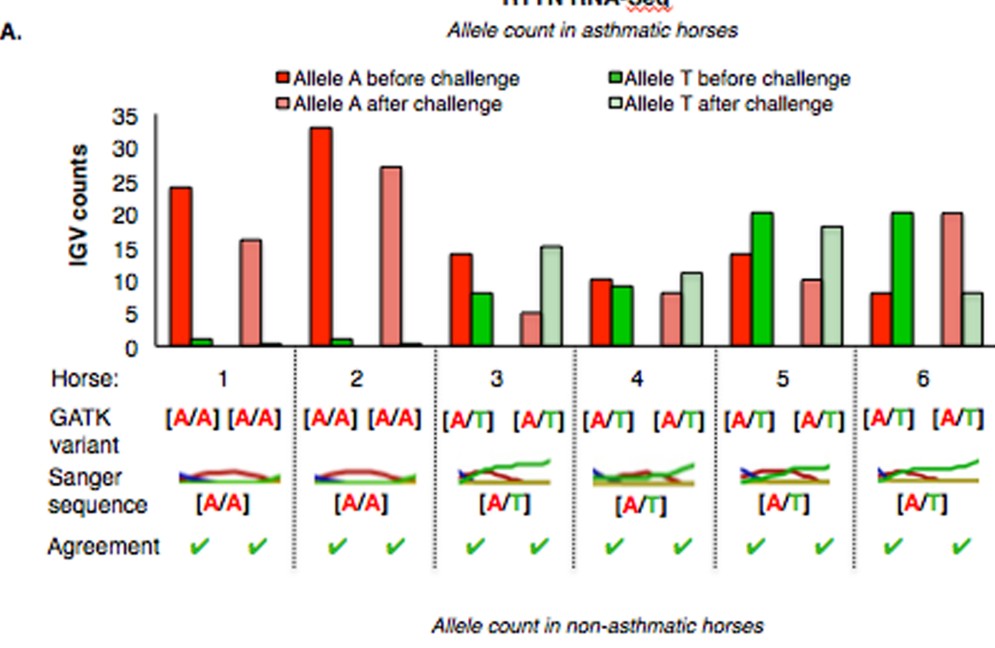

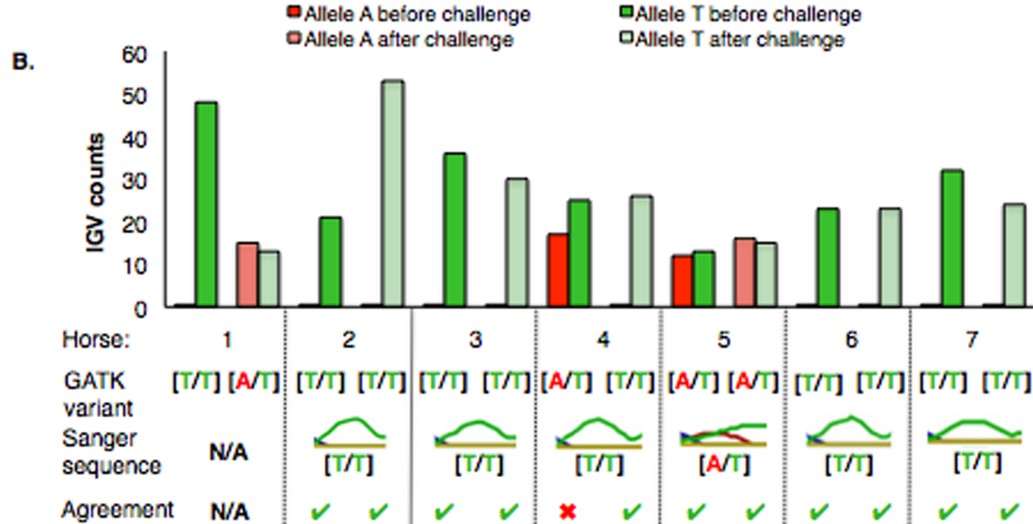

**Figure 6 Comparison of GATK variant calls and Sanger sequencing results for *RTTN* in asthmatic (A) and non-asthmatic (B) horses.** Details as in Fig. 5. (A) Four asthmatic horses (3–6) had heterozygous alleles [A/T] and two (1 and 2) were homozygous for the mutant allele [A/A]. Genotypes were consistent across horses and methods. (B) In non-asthmatic horses, one (5) had heterozygous [A/T] alleles, four horses had homozygous wild type [T/T] alleles, and two horses (1 and 4) were inconsistently identified as homozygous wild type and heterozygous in different samples. Sanger sequencing confirmed the genotype of horse 4 as heterozygous. DNA was not available for non-asthmatic horse 1.

### Sequence alignment

Sanger DNA sequences of *PACRG* from 10 asthmatic horses and 14 non-asthmatic horses (including those that were analyzed by RNA-Seq) were aligned. Among asthmatic horses, eight had the heterozygous [A/G], two had the homozygous altered [A/A] and none had the homozygous WT [G/G] genotype. Among non-asthmatic horses, six had

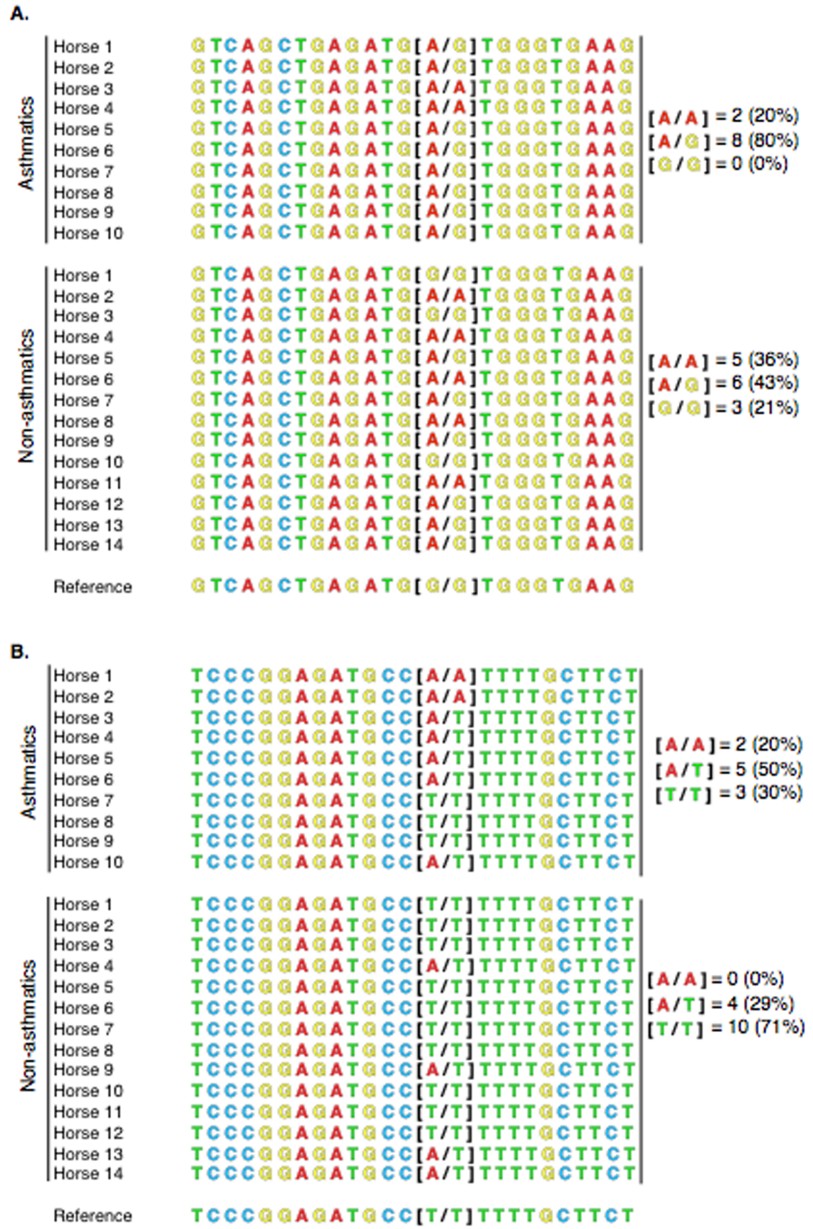

**Figure 7 Alignment of PACRG (A) and RTTN (B) Sanger sequences for 10 asthmatic and 14 non-asthmatic horses with the reference genome.** (A) For PACRG, eight asthmatic horses (80%) were heterozygous [A/G], 2 (20%) were homozygous mutant [A/A] and none was homozygous wild-type [G/G]. Six non-asthmatic horses (43%) were heterozygous [A/G], 5 (36%) were homozygous mutant [A/A] and 3 (21%) were homozygous wild type [G/G]. (B) For RTTN, in the asthmatic group, there were 5 (50%) heterozygous [A/T], 2 (20%) homozygous mutant [A/A], and 3 (30%) homozygous wild type [T/T] genotypes. In the non-asthmatic group, 4 (29%) were heterozygous [A/G], 10 (71%) were homozygous wild type [T/T], and none had the homozygous mutant [A/A] genotype.

the heterozygous [A/G], five had the homozygous altered [A/A] and three had the homozygous WT [G/G] genotype (Fig. 7A). *RTTN* alignment in asthmatic horses yielded five heterozygous [A/T], two homozygous altered [A/A], and three homozygous WT [T/T]

genotypes. In non-asthmatic horses, four had the heterozygous [A/T], 10 had the homozygous WT [T/T], and no horse had the homozygous altered [A/A] genotype (Fig. 7B).

## DISCUSSION

The goal of this study was to assess the reliability of an adapted RNA-Seq sequence variant calling workflow compared to Sanger sequencing. Sequence variant calling using RNA-Seq reads is recent practice, and reliability of results is a function of sequencing platform, depth, quality, precision of read mapping, and appropriate sequence variant calling and filtering methods. The reliability of identifying gene sequence variants using RNA-Seq has been considered uncertain. In some reports RNA-Seq was considered useful for identifying gene variants (Quinn et al., 2013; Piskol, Ramaswami & Li, 2013) while in other reports differences between RNA and DNA sequences were relatively frequent (Sheng et al., 2016; Guo et al., 2017; Li et al., 2011).

In this study we applied a modification of GATK best practices for sequence variant calling with RNA-Seq, and verified the results with Sanger sequencing. In 24 of 26 samples substitution variants in PACRG and RTTN were identified by both methods, while two horses' genotypes were discordant by RNA-Seq with inconsistent genotypes before and after challenge. Sanger sequencing confirmed one of the discordant genotypes, while the other could not be further assessed.

Two candidate substitution variants in the PACRG and RTTN coding sequence were identified after stringent filtering. Presence of the substitution variants was confirmed with PCR and Sanger sequencing in 24 samples. Correlation between RNA-Seq and Sanger sequencing showed that for PACRG both alleles of the gene were properly identified in all horses and conditions by the modified GATK workflow. For RTTN, two of the samples were misidentified by the workflow with alleles inconsistently identified before and after challenge. Lower mean expression suggesting lower sequencing coverage for RTTN might have affected the likelihood of inaccurate sequence variant calling. Nonetheless, the vast majority of alleles were identified appropriately, suggesting that the workflow is suitable for sequence variant calling in RNA-Seq at gene coverage in the 10–20 cpm range.

For supplementary analysis, SIFT was initially applied, followed by Polymorphism Phenotyping (PolyPhen) 2 and screening for non-acceptable polymorphism (SNAP) 2, to predict sequence variant effects on protein function for both substitution variants. SIFT uses phylogenetic data (Ng & Henikoff, 2001, 2002, 2003, 2006; Kumar, Henikoff & Ng, 2009), while PolyPhen2 uses structural information and multiple alignments (Adzhubei, Jordan & Sunyaev, 2013) to predict whether or not a sequence variant may cause loss of function. The two methods often yield similar results, but limited specificity suggests that results should be interpreted with caution (Flanagan, Patch & Ellard, 2010). SNAP2, on the other hand, uses evolutionary, structural, solvent-access, and annotation information, as well as data from available homologs to predict whether a sequence variant is likely to have an effect on protein function (Hecht, Bromberg & Rost, 2013; Bromberg & Rost, 2007; Hecht, Bromberg & Rost, 2015; McLaren et al., 2010).

While these three approaches can yield different results (*Dong et al., 2015*; *Thusberg & Vihinen, 2009*), inferences regarding PACRG and RTTN amino acid substitutions were consistent. However, ultimately conclusions regarding the effect of substitution variants require stringent protein functional analysis, and results from this study should be considered preliminary.

The substitutions identified changed V182M (valine to methionine) and R1807W (arginine to tryptophan) in PACRG and RTTN, respectively. For PACRG, the V->M substitution minimally affected hydrophobicity and pI, while the R->W substitution in RTTN increased hydrophobicity and decreased pI. The substitution variants were considered to potentially cause loss of function and to have non-neutral effects (Tables S1 and S2). *PACRG* is a gene conserved across species (*Thumberger et al., 2012*) that shares a bi-directional promoter with parkin (*PARK2*) (*West et al., 2003*). PACRG is affiliated with axonemal doublet microtubules, and contributes to the signaling pathway that controls dynein-driven microtubule sliding (*Thumberger et al., 2012*; *Mizuno, Dymek & Smith, 2016*; *Wilson et al., 2010*; *Ikeda, 2008*). A SNV in *PACRG* was strongly associated with an increased risk of developing childhood asthma following early-life tobacco smoke exposure (*Scholtens et al., 2014*).

For the RTTN substitution variant, tryptophan is an aromatic, non-polar, and hydrophobic amino acid often buried in hydrophobic cores, while arginine is a polar and positively charged amino acid often found on outside chains (*Betts & Russell, 2003*). *RTTN* is a cilium-associated protein (*Faisst et al., 2002*) essential for assembly of centrosomes in non-motile and motile cilia (*Chen et al., 2017*). Absence of RTTN, or presence of gene sequence variants that disrupt the interaction of RTTN with SCL/TAL1 interrupting locus (*STIL*), abrogate proper ciliary development and function (*Chen et al., 2017*), and recessive mutations in *RTTN* are linked to abnormal primary ciliary development in humans (*Kheradmand Kia et al., 2012*). A change in the structural stability or binding affinity of the entire protein or the affected residue could impact ciliary structure and function. The R1807W substitution variant in the carboxy-terminal region is not immediately proximal to the suggested centrosome-targeting and STIL-binding site (*Chen et al., 2017*) but could nevertheless result in defective centrioles and hence affect cilium structure and function.

Substitution variant sequence determination in 13 RNA-Seq and 11 additional samples showed that 80% of asthmatic animals were heterozygous and 20% were homozygous altered for PACRG, and that no individual had the homozygous WT genotype [G/G]. Conversely, among non-asthmatic animals more than half were homozygous, whether WT or altered (5 [A/A] and 3 [G/G]). For *RTTN*, 20% of asthmatic horses were homozygous altered [A/A], 30% were homozygous WT [T/T] and 50% were heterozygous. Among non-asthmatics, none was homozygous altered [A/A], while 71% of horses were homozygous WT [T/T] and 29% were heterozygous. Therefore, the substitution was present in 70% of asthmatic horses and in only 30% of non-asthmatic horses (heterozygous or homozygous altered). Albeit, the variants have been identified in only a small sample of asthmatic and non-asthmatic animals, and have to be considered as variants of unknown significance (VUS). A comprehensive genome-wide association study would be

necessary to determine association between these VUS and asthma, and statistical analysis of potential associations would need to be performed prior to filtering of variants.

Pearson's Chi-squared test with Yates' continuity correction applied detected no difference in allele frequency for *PACRG*, or in genotype frequency for *RTTN* and *PACRG*, between asthmatic and non-asthmatic horses. A significantly higher frequency of the altered allele (A) in asthmatic compared to non-asthmatic horses was identified. For *PACRG*, although not significant, the *P*-value obtained after testing for differences in genotype frequency ($P = 0.213$) was lower than when testing for allele frequency ($P = 1$). This finding may be attributed to the higher proportion of asthmatic horses with a heterogeneous genotype (WT/alt for eight of 10 horses) compared to non-asthmatics (WT/alt for six of 14 horses). However, changes in allele frequency and potential roles in the pathogenesis of asthma are of unknown significance due to the small sample size in this study. Notwithstanding, a significant difference in the frequency of the *PACRG* heterozygous genotype has been reported in pulmonary tuberculosis in humans (*Udina et al., 2007*). A genome-wide interaction study also identified a *PACRG* SNP to be linked to an increased risk of developing childhood-onset asthma following early-life exposure to tobacco smoke (*Scholtens et al., 2014*). SNPs in *PACRG* also contributed to susceptibility to tuberculosis (*Bragina et al., 2016*).

For *RTTN*, the difference in allele frequency was encouraging and further analysis with a larger number of samples to assess association with asthma may be warranted. RTTN is a centrosome-associated protein first discovered for its role in axial rotation and left–right specification in the mouse embryo (*Faisst et al., 2002*). The R->W substitution altered the hydrophobicity and isoelectric point at position 1,807, and R <=> W substitutions were predicted to be most problematic in the human genome (*Majewski & Ott, 2003*). In addition, R->W substitution is generally disfavored in all protein types (*Betts & Russell, 2003*). Overrepresentation of mutated arginine was a prominent feature among disease-causing mutations in a range of conditions (*Khan & Vihinen, 2007*).

As landmarks in epithelial-environmental interaction, cilia are highly specialized cellular projections. Most vertebrate cells have a single non-motile ("primary") cilium that transduces signals from the environment or other cells, while motile cilia occur in multiples on specialized cells of the respiratory tract, oviduct and ventricles of the brain (*Reiter & Leroux, 2017*; *Szymanska & Johnson, 2012*). Motile cilia directionally propel cells or extracellular fluid through "metachronal wave" beating movements (*Eshel & Priel, 1987*; *Gheber & Priel, 1990*). The ability of motile cilia to beat in a synchronized manner requires specialized proteins that are absent in non-motile primary cilia, but otherwise both types of cilia have similar internal architecture. The main part of the cilium is the axoneme, which is comprised of nine outer microtubule doublets, one central microtubule pair (in motile multiple cilia) and a multitude of affiliated proteins. Prominent among these are tektins that stabilize microtubules and regulate axoneme length (*Steffen & Linck, 1988*), and protofilament ribbon proteins that are essential for sliding of adjacent microtubule doublets to generate ciliary movement (*Linck & Norrander, 2003*). Abnormalities in cilia are now appreciated as cause for the development of respiratory diseases, often through gene sequence variants associated with a loss of function

affecting unique ciliary proteins (*Reiter & Leroux, 2017*). Ultrastructural changes were previously reported in the ciliated epithelium of horses with severe asthma (formerly called chronic obstructive pulmonary disease), and included loss of ciliated cells (*Kaup, Drommer & Deegen, 1990*). Factors that affect beating, synchronization or orientation of motile cilia result in accumulation of mucus in airways (*Reiter & Leroux, 2017*), which is a prominent feature of equine asthma. Furthermore, hedgehog (HH) signaling is strongly linked to ciliary function, and many components of the HH signaling pathway localize to cilia (*Huangfu et al., 2003*; *Goetz & Anderson, 2010*). However, considering the relatively small number of individuals tested, allele frequencies identified in this manuscript, and their potential impact on ciliary function, remain to be confirmed on a larger scale.

PACRG may be linked to HH signaling in mice where patched1 (*PTCH1*) and *PACRG-PARK2* loci are thought to interact and regulate ciliary function in ependymal cells (*Gavino & Richard, 2011*). Interestingly, *PTCH1* is differentially expressed in asthmatic compared to non-asthmatic horses following challenge (*Tessier et al., 2017*), linking *PACRG* and an asthmatic response to environmental agents with the HH pathway. The PACRG protein associates with protofilaments (*Ikeda et al., 2007*) of the ciliary axoneme (*Thumberger et al., 2012*; *Lechtreck et al., 2009*; *Dawe et al., 2005*), has a role in ciliary morphogenesis and function (*Wilson et al., 2010*) and is directly involved in ciliary motility through control of dynein-driven microtubule sliding (*Mizuno, Dymek & Smith, 2016*). PACRG also has a variety of interacting partners such as microtubules, α- and β-tubulin and meiosis/spermiogenesis associated 1 (MEIG1) protein, heat shock protein (HSP) 70 and HSP 90 (*Ikeda, 2008*; *Li et al., 2015*; *Imai et al., 2003*). Impaired function or interaction of PACRG with its partners could weaken or impair ciliary stability and motility. The exact nature and function of methionine in protein structure remains incompletely understood, and substitutions involving methionine has been associated with several diseases (*Valley et al., 2012*). Both valine and methionine are hydrophobic residues grouped among the least polar amino acids (*Wolfenden, 2007*). Methionine is a sulfur-containing amino acid that is among the most hydrophobic residues and also easily oxidized if exposed (*Brosnan & Brosnan, 2006*). Although V->M substitutions are generally neutral, methionine's sulfur connected to a methyl group would make it less likely to interact with other proteins (*Betts & Russell, 2003*). Methionine was overrepresented as a mutant residue in several mutations associated with decrease or loss of function (*Khan & Vihinen, 2007*), including the human androgen receptor (*Kazemi-Esfarjani et al., 1993*). Although the effect of a V->M substitution is unknown, any change in PACRG structure or binding affinity could impact ciliary function, and may be of great interest in the context of severe asthma.

## CONCLUSIONS

Sequence variants can be confidently called with RNA-Seq, although the required minimal coverage remains to be clearly defined and may be variable. Single point substitution variants in *PACRG* and *RTTN* were detected in all asthmatic horses, and although there was no significant difference in allele and genotype proportions between the two groups, the altered allele in the RTTN gene was more prevalent in asthmatic compared to

non-asthmatic horses. Functional cilia are crucial for lung health, and sequence variants resulting in impaired protein function are likely to have a negative impact. The significance of the substitutions in *PACRG* and *RTTN* remains to be determined but they are of potential interest for future investigations.

# ACKNOWLEDGEMENTS

The authors thank Laurent Viel, Mary Ellen Clark and Andrés Diaz-Méndez for assistance with animal handling, biopsy collection, and respiratory function measurements.

## Funding

This work was supported by Equine Guelph and the Ontario Ministry of Agriculture and Rural Affairs (no. 051644), and a doctoral scholarship to L. Tessier from the Ontario Veterinary College. The funders had no role in study design, data collection and analysis, decision to publish, or preparation of the manuscript.

## Grant Disclosures

The following grant information was disclosed by the authors:
The Ontario Ministry of Agriculture and Rural Affairs: 051644.
The Ontario Veterinary College.

## Competing Interests

Olivier Côté is employed by BioAssay Works LLC, Ijamsville, MD, USA. Laurence Tessier became an employee of BenchSci, Toronto, ON after completion of this study. Dorothee Bienzle is an Academic Editor for PeerJ.

## Author Contributions

- Laurence Tessier conceived and designed the experiments, performed the experiments, analyzed the data, contributed reagents/materials/analysis tools, prepared figures and/or tables, authored or reviewed drafts of the paper, approved the final draft.
- Olivier Côté conceived and designed the experiments, performed the experiments, contributed reagents/materials/analysis tools, authored or reviewed drafts of the paper, approved the final draft.
- Dorothee Bienzle conceived and designed the experiments, performed the experiments, analyzed the data, contributed reagents/materials/analysis tools, authored or reviewed drafts of the paper, approved the final draft.

## Animal Ethics

The following information was supplied relating to ethical approvals (i.e., approving body and any reference numbers):

All procedures were approved by the Institutional Animal Care Committee of the University of Guelph (protocol R10-031) and conducted in compliance with the Canadian Council on Animal Care guidelines.

## Data Availability

The raw sequences are publicly available under study PRJNA384774 (SRP106023).

## Supplemental Information

Supplemental information for this article can be found online at http://dx.doi.org/10.7717/peerj.5759#supplemental-information.

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
