# Peer review of "Sequence variant analysis of RNA sequences in severe equine asthma"

_PeerJ, doi:10.7717/peerj.5759_

## Round 0.1 · original submission · Major Revisions

The reviewers have made thoughtful comments on methodology, interpretation, and presentation. Please address each comment if you choose to submit a revised manuscript.

Reviewer 1 ·

Basic reporting

The manuscript is well written, however this manuscript could benefit from following HGVS nomenclature (http://varnomen.hgvs.org/). In the manuscript the word 'variant' is used both to represent a variable position in the genome (a SNP) as well as to represent a specific base (A,T,C,G) at a SNP. This should be clarified.

Experimental design

The methods regarding alignment, variant calling, assessment of protein function, and validation with Sanger sequencing are well done. Disclaimer: I am not an expert with the programs used for assessing protein function.

The experimental design to associate genetic variants and gene expression to the asthma phenotype in this paper is very poor. The authors do not perform any statistical tests to associate the identified genetic variants or gene expression to asthma.

Validity of the findings

no comment

Additional comments

General Comments:
The manuscript titled "Variant analysis of RNA sequences in severe equine asthma" by Tessier et al. aligned RNAseq data with STAR, called and filtered DNA sequence variants with GATK and VEP, identified novel variants with VEP, assessed each variants' impact on protein function with PolyPhen2 and SNAP2, and argue that two of these variants impact protein function which are also relevant in asthmatic horses. The objective was to assess the reliability of calling variants from RNAseq data. Reliability was determined by comparing genotypes called by GATK from the RNAseq data to Sanger sequenced data. The methods regarding alignment, variant calling, assessment of protein function, and validation with Sanger sequencing are well done. Therefore the main objective of the paper is well done. Notably, the version of GATK used (v3.2.2) is years out of date (a more recent v3.3 was released in 2014), however re-analysis with v3.7 or v4 is not required.
However, the paper is severely hampered by attempting to find a connection of DNA variants and RNA gene expression to asthma disease status. No statistical tests were performed in this manuscript to quantify if gene expression or DNA variants are relevant to asthma. The authors never state that they have a significant association nor blatantly misrepresent their findings. However, by focusing on two specific variants (identified after a procedure to identify "variants of interest") that have altered genotype frequencies between 10 asthmatic and 14 non-asthmatic individuals. They consistently imply that these variants are relevant to the asthmatic phenotype throughout the manuscript when no statistical tests have been performed. For example: in the conclusions the authors write "The heterozygous mutant genotype of PACRG was more prevalent among asthmatics while the homozygous-WT genotype of RTTN was more prevalent in non-asthmatics," and in Results they write "Expression of PACRG overall was slightly higher in asthmatic than non-asthmatic horses." Presenting these data in such a manner could be very misleading to non-experts that read this manuscript who might not realize that no statistical association to asthma was found or even performed.
Regarding DNA variants and asthma:
Asthma is a complex trait which likely has many small effect loci that effect disease susceptibility or progression. This is why in horses GWAS for SNPs associated with severe equine asthma with ~370 individuals (~350k marker density) or ~650 individuals (~50k marker density) are still underpowered to find a single variant that is significantly associated to asthma after genome wide multiple testing correction (Shakhsi-Niaei et al. 2012, Schnider et al. 2017). For reference, meta-analyses in humans are reaching sample sizes in the tens of thousands (and even hundreds of thousands) to find significant associations (Ferreira et al. 2014 doi:10.1016/j.jaci.2013.10.030, Zhu et al. 2017 doi: http://dx.doi.org/10.1101/133322). With 24 samples the likelihood that the authors found a significant association after multiple testing correction is very low.
1. Specifically – Results:
a. Although important to present in this paper, the estimated genotype and allele frequencies from 10 asthmatics and 14 controls are likely unstable within each group because the sample sizes (10 and 14) are too small to accurately represent population frequencies.
2. Specifically – Methods Lines 176-195:
a. They describe a procedure to identify variants of interest (i.e. variants that impact protein function and are relevant to asthma).
b. This section is supposed to be a GWAS to test for how well any particular SNP predicts asthmatic disease status (to identify variants of interest and relevant to asthma). They should test all potentially impactful high confidence protein modifying variants for association to asthma in a linear model (logistic regression, DiseaseStatus ~ Genotype) and correct for multiple testing by FDR or Bonferroni correction (example α/NumberOfComparisons).
i. Notably to help reduce multiple testing correction burden they could remove variants violating HWE in control individuals, use a MAF > 0.05, etc.
c. The authors 'by-pass' the multiple correction burden and identify 'variants of interest' essentially by filtering for variants "present in all asthmatic horses but not all non-asthmatics".
d. This procedure is statistically unsound and does not statistically link any variant to asthma.
In regards to RNA sequencing and asthma:
There are some very strange sentences in this manuscript that attempt to support differential expression (DE) results from one of their other manuscripts [56] that found the majority of genes (81 of 111) to be upregulated. Example, in the Results lines 219-222 and specifically in the Discussion lines 309-313: the authors attempt to summarize the expression of all ~20,000 protein coding genes as 'slightly higher' in 'some' asthmatic horses after challenge by visual inspection of aligned sequences in IGV.
1. This is unacceptably subjective and is not a result that can support their previous paper.
a. The authors recognize that this procedure had 'limited precision', however this is not enough to justify its presence in the paper.
2. At best, viewing the number of aligned RNA sequences would represent the raw counts. However, raw counts are never analyzed for differential expression; all counts must be normalized (usually by the library size) before one can make a conclusion on the biological implications. With raw counts the observed differences in sequence depth could simply be due to more sequences being sequenced in some individuals.
3. The authors need to perform a DE analysis (GeneExpression ~ DiseaseStatus) with normalized counts before attempting to support their other paper.
The authors present unacceptable connections between gene expression in the two 'interesting' genes PACRG and RTTN in the Results section lines 249-254.
1. Here they present a range (min-max) of counts-per-million (CPM, a normalized RNAseq count) counts in asthmatics and non–asthmatics.
2. Minimally, the authors must provide mean and/or median CPM counts per gene in each group so the reader can imagine the distribution.
3. However, again the final sentence states that "Expression of PACRG overall was slightly higher in asthmatic than non-asthmatic horses."
a. Again, although not explicitly stated this sentence is clearly present to make the reader think that there is a difference (and specifically upregulation of genes genome wide) between asthmatic and non-asthmatic horses when no statistical analysis (DE, or a t-test) was performed, and the range of CPM counts for asthmatic and non-asthmatic horses are almost perfectly overlapping anyways (asthma 30.7-66.3 and non-asthmatics 25.1-65.6). Are the means different? Is there differential expression?
Specific comments:
1. Abstract - Results:
a. Line 61-63: What are the frequencies for the PACRG heterozygous genotype, and RTTN homozygous genotype in asthmatics compared to controls?
b. For RTTN the authors write "homozygous mutant phenotype". I believe they meant genotype. A phenotype cannot be homozygous. If the genotype determines the phenotype, please refer to the increased frequency of the genotype.
c. Minot point, but important.
i. Use of "mutant" should be avoided when referring to DNA variants. Nomenclature (at least for humans) regarding DNA sequence variants has been standardized and I recommend that the authors follow guidelines at: http://varnomen.hgvs.org/ . More specifically: http://varnomen.hgvs.org/bg-material/basics/ under the Mutation and polymorphism section.
1. "In some disciplines the term “mutation” is used to indicate “a change” while in other disciplines it is used to indicate “a disease-causing change”."
2. It is generally preferred to avoid terms like "mutation" and use more neutral terms like “sequence variant”, “alteration” and “allelic variant”.
ii. for examples see http://varnomen.hgvs.org/bg-material/simple/ .
iii. The authors are describing a deleterious effect on function at the protein not DNA level.
2. Results – RNA-Seq coverage:
a. Lines 219-221: Remove from results section "With the IGV tool large-scale genomic data sets are visualized in real-time over a wide range of resolutions [72]."
i. Not a result
b. Line 221-222: After visual inspection the number of aligned RNAseq reads were determined to be "slightly higher" in "some" asthmatic horses following challenge.
i. What was the value of the "similar" coverage across horses and challenges?
ii. The change in depth of coverage needs to be quantified.
iii. How many asthmatics?
iv. Is this result relevant to the conclusions of this paper?
3. Results – Variant calling and filtering:
a. Lines 227-228: which variants and what is 'lowest prevalence'. Please clarify reference/non-reference and which base, and quantify frequency.
b. This section would benefit by following HGVS nomenclature.
4. Results – Protein alignment:
a. Specify "Amino acid sequence alignment" in tittle of this section.
5. Results – Sanger sequencing:
a. Should this section be titled "Confirmation of RNAseq with Sanger sequencing"?
b. I bring this up because the total number of individuals is different if the reader thinks the authors are referring only to the Sanger sequencing.
c. Line 267: please specify the number of asthmatics and non-asthmatic horses.
6. Results – Expression of PACRG and RTTN:
a. Why is it relevant to differentiate cpm values in asthmatic and non-asthamtics when no differential expression analysis is performed? Or should we trust the genotype calls more in asthmatic vs. non-asthmatic?
b. What is the mean/median expression in cpm for asthmatics and non-asthmatics?
c. Lines 253-254: This sentence needs to be removed or clarified with some statistical analysis (see below).
d. Although not explicitly stated, this sentence implies that there is differential expression and upregulation of PACRG in asthmatic horses. A differential expression analysis or t-test must be applied for this sentence to remain in the paper.
e. Asthmatics have expression of 30.7-66.3 and non-asthmatics have expression values 25.1-65.6. With the information provided these distributions seem to be nearly perfectly overlapping. I see no support for the last sentence (lines 235-254).
f. Was this gene upregulated in their other study [56]?
7. Results – Sequence alignment:
a. This section presents the evidence (genotype frequency imbalances between case and control individuals) for the connection of these two DNA variants to the asthma phenotype.
b. Unfortunately, no statistical test is performed to quantify how well these variants predict disease status.
i. Specifically, this section would greatly benefit from logistic regressions DiseaseStatus ~ Genotype. One for each variant. Importantly, these regressions should not only be performed on the 2 variants discussed but ALL potentially impactful high confidence protein modifying variants. The exact number of variants that should be tested is not listed in the methods section however, from EVP the authors list 26619, 24527, 28909, and 28451
ii. Additionally, to do this in a more appropriate manner I would require genotype to be a significant predictor of DiseaseStatus at α < 0.005. As this represents a bonferroni correction for 10 independent tests.
c.
8. Discussion:
a. Line 310-312: Again, summarizing the depth of coverage for aligned reads for 20,000 protein across multiple individuals like this is inappropriate. These very subjective sentences should be quantified or removed.
i. Quantifying the depth of coverage would be beneficial if attempting to support variant or genotype calls.
ii. However, given the next sentence (312-313) it seems like the authors are attempting to state that genes are upregulated in asthmatics (after visual inspection of aligned reads). Which would support their other paper [56]. However, differential expression analyses are only done after normalization of counts. Therefore, observing the raw counts might just mean that more sequences were sequenced in 'some' asthmatics and have no biological meaning. This sentence does not support a conclusion on differential expression.
b. Line 312-313: When considering the previous sentence and this sentence together. It seems as though the authors are attempting to state that the majority of differentially expressed genes (81 of 111) were previously found to be upregulated as stated in reference [56].
i. "prominent changes" doesn't mean upregulated.

Annotated reviews are not available for download in order to protect the identity of reviewers who chose to remain anonymous.

·

Basic reporting

no comment

Experimental design

Major comments:

1. What was the degree of relatedness of the horses in the institutional research herd. if there was a degree of relatedness then this may reduce the likelihood of identifying variants related to disease.

2. The authors state that "10 were missense mutations, coded for proteins and had SIFT scores <0.01" It would be better if all 10 variants were identified in a supplemental table as they only go on to discuss the PARCG and RTTN variants only.


Minor comments:
1. Sequence coverage. the authors state that there was "slightly higher coverage in some asthmatic horses following challenge” - were the genetics regions only adequately covered for variant calling in the asthmatic challenge samples? If so were these trimmed from the data set before comparison with non-asthmatic samples? Otherwise this would give flase positive as SNPs would be more likely to be found in asthmatics due to better coverage.

2. Was the Sanger sequencing carried out on cDNA or gDNA? specify

Validity of the findings

While the variants identified in PARCG and RTTN are of interest and affect plausible biological pathways in relation to airways disease, I feel the the authors should explicitly note in their discussion that there was no significant difference in genotype frequency between the asthmatic and non-asthmatic horse groups.

Additional comments

This is a well written paper that utilises established bioinformatic methods to identify coding variants from RNA seq data. The potential significance of variants in 2 genes, PARCG and RTTN, are analysed using in silico methods and genotypes were verified using Sanger sequencing. The significance of the observation is limited by the small group sizes and I feel that fact that there is no difference in variant alley frequency between groups should be explicitly acknowledged in the discussion and conclusions (e.g. lines 403 - 404) as noted above.

·

Basic reporting

This article is clearly written and well-referenced.

Some of the figures are not needed (see specific comments below).

No hypothesis was stated. it seems that the authors have incompletely analyzed their data relative to their objective.

Experimental design

I have several questions and comments relative to the experimental design, and I believe that this is a potentially powerful data set that should be re-analyzed and in some places additional experimentation done and additional data [resented (see specific comments below).

Validity of the findings

I believe that some of the authors findings are not valid and the study suffers from inadequate data quality controls and lack of statistical analysis (more detailed comments below).

Additional comments

ABSTRACT

Lines 36-37: The author’s statement--- “The disease in horses has complex inheritance
37 including both dominant and recessive patterns that are ill defined.” ---is contradictory and incorrect based on the current literature, one can conclude that the inheritance is complex, but there is not good evidence for either dominant or recessive patterns of inheritance for RAO.

INTRODUCTION

Lines 93-120 seemed to be more appropriate for the discussion than the introduction; in particular since there is not equine data that I am aware of that directly implicates cilliary dysfunction in RAO, and because the author’s objective was not to evaluate cilliary function or these two genes at the start of this research.

METHODS

Line 132-133 It is clear that the control horses were exposed and biopsies taken after 3 days of exposure, but it is unclear if the asthmatic cases were biopsied when clinical signs appear (i.e. 1-3 days) or at 3 days. Please clarify.

There are several places in the methods where results are reported. These data should be moved to the results section. For example:

Line 181: Suppl. Figure 1

Lines 182-188: “In asthmatic horses, a total of 26,619 pre- and 24,527 post-challenge variants were identified, respectively, while the corresponding numbers were 28,909 and 28,451 for non-asthmatic horses. Approximately 30% of variants were novel and not previously described. The types of variants and their coding region effects are summarized in Suppl. Figures 2 and 3. For further variant selection using VEP the inclusion criteria were 1) missense mutation in protein-coding sequence; and 2) predicted to cause loss of protein function.”

Lines 191-194: “Ten variants present in all asthmatic horses (before or/and after challenge) but not all non-asthmatic horses were identified. Of these, only two variants were expressed in all samples and were more prevalent in asthmatic compared to non-asthmatics.”

Line 194: Why would the authors not evaluate the putative functional effect of all discovered variants---this seems like a much better approach, in particular since their sample size is too small to clearly delineated differences in variant frequencies especially with the multiple testing burden with thousands of variants identified.

Line 203: The authors use Sanger sequencing of bronchial cDNA to “confirm” variants identified by RNAseq and to genotype the horses. However, this is not the appropriate approach. The SNVs that are detected should be present in genomic DNA and genomic DNA would be the appropriate reference material to confirm genotype. Unlike experiments that are interested in identifying novel somatic mutations in RNAseq (which is where variants calling in RNAseq originated); these authors are using RNAseq to identify functional genomic variation (instead of doing whole genome sequencing that is the most appropriate, but is more expensive option). To conform what was found in the cDNA genomic DNA is most appropriate for this application--unless you were looking to confirm novel splice variants---which unfortunately the authors failed to do.

No statistical analysis appears to have been performed.

COMMENTS ON THE RESULTS PRESENTED IN THE METHODS

Lines 182-188: Because, the authors were not looking, nor should they expect, novel somatic mutations in RAO (as stated above), it is very disconcerting to find huge differences in the variants detected in the cases (or controls) before and after challenge. To the reviewer this suggests one of three issues with the dataset:

1. The data quality was insufficient to accurately call variants resulting in widely different variant calls across the two time points. Unfortunately the reviewer cannot determine if this is the case as the authors fail to report sufficient information regarding the RNAseq data and its quality, or the variant quality scores of their cut-offs used (missing form methods) this cannot be assessed (see below).
2. The differences are due to the presence or absence of particular genes in the dataset at the different time points. Again without information about the read depth, number of expressed genes or the overlap in the expressed genes between each time point, this cannot be assessed.
3. The read depth was insufficient to accurately capture low abundance reads resulting in different genes being captured in each library.

RESULTS

Lines 219-222: The authors do not report sufficient information regarding the RNAseq data, for example:
1. The total number of reads collected per sample or minimally mean/median and range.
2. The average mapped read depth and coverage. This should be calculated; figure 1 is not useful or adequate to determine the utility of the data.
3. The total number of genes identified in samples from each time point and how those genes overlap across case/control and time point which would be important to report to address #2 above.
4. An indication of read quality.

Lines 255-226: the authors need to report variant quality scores and the cut-off used. The reviewer suspects that the large differences in variants called are do to poor quality reads and/or insufficient quality control. In its current format supplemental figure 1 is not that useful. A Venn diagram of the overlap in genes detected across the samples would be more useful.

Lines 227-228: These data should not be reported a “prevalence.” Allele frequencies should be reported and allele frequency differences should be calculated across all the variants and the authors should use a statistical analysis to come to the conclusion that allele frequencies are different. Authors should consult a statistician to perform statistical analyses of their data. The authors also need to be aware of correction for multiple testing as they consider allele frequency differences in the data.

Lines 224-233: Why did the authors not evaluate other changes in the data besides SNPs? RNAseq give a great opportunity to look a splice variants and it’s disappointing that this was not done.

Lines 249-254: The methods do not adequately describe the gene expression differences and the analysis that led to these results. Where the data normalized? What was done to correct for multiple testing, what where the p-values or q-values, what was the FDR? It would be more useful to have the mean and SD (and/or fold-change) for the expression in the groups and have some sort of statistical reliability associated with the results.

Lines 266-280: See comments above authors need to do comparisons to genomic DNA.

Lines 284-287: As stated before the authors should present allele frequencies (as this is a complex trait. PACRG allele frequencies are not statistically different between cases and controls: Cases G=0.4, A=0.6, and controls G=0.43 and A=0.57.

Lines 289-292: Allele frequencies should be reported and a statistical test performed. This test should also account for multiple testing given the 100s of variants identified (and ideally actually tested for differences).

DISCUSSION

Lines 297-300: This statement is true, so why did the authors not report depth, quality, read mapping quality, variant quality and make appropriate cut-offs in their data?

Line 309-312: IGV viewer not adequate and the authors cannot make conclusions about similarity of differences in expression based on this method.

Lines 332-334: The authors need to go back and correct for these omissions in their data analysis.

Lines 349-354: These comments about the PACRG alleles are not appropriate as the frequencies are not statistically different.
Lines 360-361: Adding a statistical test would allow the authors to make a true statement here.

Lines 402: The authors have not demonstrated this statement with their data.

---

## Round 0.2 · Major Revisions

As you can see, Reviewer 1 continues to have major concerns, while Reviewer 2 thinks the manuscript has been appropriately revised for the most part. If you can address the concerns of Reviewer 1, I am willing to consider the manuscript one more time. If you choose to resubmit, I will consider whether you have adequately addressed the concerns, without sending the manuscript out for further review. If the concerns have not been adequately addressed on the third submission, it is highly unlikely the manuscript would be evaluated a fourth time.

The comments of Reviewer 1 have been attached as a PDF for your convenience. As you see, the reviewer is primarily concerned that you are ascribing association when rigorous statistical treatment probably warrants only a trend. As both reviewers note, and you state, the work is primarily a test of the utility of RNA-Seq to call gene sequence variants and identify potential disease association. You have toned down the claims for disease association, but not enough. You have done a considerable amount of work, but at this point, a major value of the manuscript, should it be published, might be as a cautionary tale of the importance of having adequate power and using appropriate statistical techniques to avoid false associations. It would be reasonable to identify trends towards association with one or two variants, but methodological rigor is more important than promoting a possible association at this point.

Reviewer 1 ·

Basic reporting

I mentioned in my last review that the authors consistently imply that there is an association between the two variants of interest and asthma when no statistical tests were performed.
1. In some ways this has been addressed. However, there are some notable exceptions.
2. Example:
a. No association was found for allele or genotype frequency of the variant in the PACRG gene and asthma (p=1.0 and p=0.2).
b. However, the authors still suggest that with a larger sample of individuals "the difference in frequency of the PACRG heterozygous genotype between asthmatic and non-asthmatic horses may increase."
c. I mentioned last time that the allele and/or genotype frequencies were likely not stable with the sample sizes reported. Why do the authors report a trend to a future experiment when the statistical test is not significant, and the number it is based on might not be stable?
d. Sentences like this need to be removed or changed.

Experimental design

The purpose of this study was "to determine the utility of RNA-Seq to call gene sequence variants, and to identify sequence variants potentially associated with asthma". I will focus my review on these two aspects.
1. the accuracy of RNA-Seq data to determine DNA genotypes
2. the variant enrichment procedure and their association to asthma
We will focus on the second aspect in this section:
During the last revision I focused on the lack of statistical power and statistical tests to determine if SNPs or RNA expression was associated to asthma.
1. Now, the authors have performed statistical tests to determine the strength of association for (DNA) altered allele frequencies and altered genotype frequencies to asthma for two SNPs of interest (one in PACRG and one in RTTN).
2. Although these statistical tests were performed, it is difficult to trust the authors variant enrichment procedure to select these two variants.
3. Additionally, multiple testing correction was not considered as previously requested.
a. The variant enrichment procedure (for variants of interest) starts by using 26,619 and 24,527 variants pre- and post- challenge in asthmatic horses and, 28,909 and 28,451 variants pre- and post- challenge in non-asthmatic horses.
b. Of these variants they called "consensus sequence variants" within each group pre-asthma, post-asthma, pre-non-asthma, post-non-asthma, were called with SeqMule.
c. Now I assume that they filtered the consensus variants per group to only consensus sequence variants unique to the asthmatic group pre- (2823) and post- (1788) challenge.
i. This step is only mentioned in the Result, line 213: "The GATK variant calling and filtering workflow yielded 2823 and 1788 sequence variants present specifically in the asthmatic group pre- and post-challenge, respectively (Suppl. Figure 1)."
1. Please add this to the methods (around line 165 or so) and clarify if these numbers were from consensus sequence variants per group, when exactly this filter was applied, and if the variants needed to be unique to each group.
ii. If this step was performed, this by-passed much of the multiple testing correction burden.
iii. By calling consensus sequence variants (I'm still not sure if they are calling consensus nucleotide or consensus genotypes) they are calling variants that are the most frequent in each of the four groups. Then the authors only use consensus variants that are unique (I think unique because they made a Venn diagram, this is not explicitly stated) to asthmatic pre- and post- groups. This filter enriches for variants that will be associated with their desired alternative hypothesis.
iv. The authors should not enrich for SNPs that will agree with the their alternative hypothesis prior to statistical analyses. Example: in a GWAS researches do not enrich for SNPs associated with their dependent variable prior to the analysis to reduce the multiple testing correction burden.
v. However, they could filter variants based on detrimental protein effects (the next step) prior to statistical analyses.
d. Then I guess the authors use VEP and SIFT to identify variants that might affect protein structure. However, they do not state how many impactful variants are removed or kept by this filter. However, I assume that it is >=10 but <= 4611 (2823+1788).
e. After this step the authors enrich for variants present in all asthmatic individuals and not present in all non-asthmatic individuals which leaves them with 10 variants (again enriching for variants that will agree with the alternative hypothesis). Then by 'manual verification' 8 of the 10 variants are excluded and they are left with their two variants of interest in PACRG and RTTN.
i. How are 80% of these variants excluded? Were these 8 SNPs genotyped incorrectly? This is disconcerting.
The variant enrichment procedure would greatly benefit from a flow chart depicting methods. Did I misinterpret these steps?
At a minimum the authors should have performed a statistical association with the asthma phenotype for some X number of SNPs, followed by multiple testing correction for these X comparisons, as mentioned in my previous revision.
This is important as the authors results conclude that there is a significant association between altered allele frequencies for the variant in RTTN p = 0.042 and asthma. However, after a Bonferonni correction for multiple comparisons of only two tests (required alpha = 0.05/2 = 0.025) this is no longer considered significant. Notably, in the current version of the paper two tests were performed for the two variants of interest.

Validity of the findings

The authors only validate two of the RNA-seq SNPs with Sanger sequencing (in ~24 individuals). Since this is a main objective of the paper I expected more than two of the ~35,000 variants (I summed the numbers from the Venn diagram) confirmed. Therefore accuracy measurements of the RNAseq data to call DNA genotypes/SNPs is questionable.
Although it is good that the authors verified these two variants of interest with sanger sequencing.
I worry that the by enriching for variants that will tend to agree with their alternative hypothesis prior to testing improperly bypasses the multiple testing correction burden and is leading the authors, and potentially others to be more interested in these two variants than they should be. I am worried that the lack of statistical association to asthma is being disregarded to some extent, and therefore the relevance of asthma to these two variants is being inflated. If they state that there is a significant association to asthma for RTTN, then they should be held to the same statistical standards of other groups. Of course, it is possible that these variants are relevant to asthma, however, it seems that the statistics do not strongly support this conclusion when considering multiple testing.

Additional comments

The genotypes were called for 26,619 and 24,527 variants pre- and post- challenge in asthmatic horses, and 28,909 and 28,451 variants pre- and post- challenge in non-asthmatic horses.
1. Line 165: The authors conclude that the "Difference in sequence variants before and after challenge were likely attributable to difference in gene expression, as previously discussed (39)."
2. Why would the difference in the number of SNPs pre- and post- challenge be discussed in their differential expression paper cited? I did not find any variants called in citation 39.
3. The difference in variants identified pre- and post- challenge are likely due to differences in alternative splicing, allele-specific expression, or simply a change in gene expression due to a time. The total quantity of RNA molecules expressed (as discussed in citation 39) is somewhat irrelevant when trying to justify the difference in variants called pre- and post- challenge. Rather, the total number of RNA molecules expressed can change due to allele specific expression etc.
4. I don't believe this was appropriately presented in the text. This should be discussed as a limitation.
5. Another thought, the difference in the number of sequence variants shown in the Venn diagram is likely very misleading if I interpret the methods correctly. They called "consensus sequence variants per group" and therefore I assume the most frequent (genotype or base-pair) was selected per group. This does not mean that the sequence variants were absent from the other groups, but rather that it was not the most frequent. Therefore, by choosing the most frequent (consensus) variants per group they might be inflating the differences between groups. Is this correct?

Line 172-173: what are existing sequence variants?
"existing sequence variants were excluded"

What was the minimal depth required for calling variants?

Please consider adding a dedicated limitations section.

Annotated reviews are not available for download in order to protect the identity of reviewers who chose to remain anonymous.

·

Basic reporting

The revised manuscript is, as previously, clearly presented.

Experimental design

The experimental design has been improved in response to the reviewer's comments.

Validity of the findings

The main criticism of the previous reviewers was the over interpretation of the results of the variant identification with respect to asthma, give the clear lack of statistical significance in variant frequency between groups. Throughout the revised manuscript the authors have toned down the conclusions in line with this.

The paper in now more appropriately focussed as a proof of principle for using RNAseq in horses to identify variants for future studies of the genetics of asthma and other conditions.

While this limits the significance of the results presented to the research community, it improve the technical accuracy of the manuscript.

---

## Round 0.3 · accepted · Accept

You have responded appropriately to the concerns of the reviewer. Best wishes for your future work.

#